# Atomic Layer Etching Using a Novel Radical Generation Module

**DOI:** 10.3390/ma16103611

**Published:** 2023-05-09

**Authors:** Junho Jung, Kyongnam Kim

**Affiliations:** Department of Energy & Advanced Materials Engineering, Daejeon University, Daejeon 34520, Republic of Korea

**Keywords:** atomic layer etching, semiconductor, EPC, plasma

## Abstract

To fabricate miniature semiconductors of 10 nm or less, various process technologies have reached their physical limits, and new process technologies for miniaturization are required. In the etching process, problems such as surface damage and profile distortion have been reported during etching using conventional plasma. Therefore, several studies have reported novel etching techniques such as atomic layer etching (ALE). In this study, a new type of adsorption module, called the radical generation module, was developed and applied in the ALE process. Using this module, the adsorption time could be reduced to 5 s. Moreover, the reproducibility of the process was verified and an etch per cycle of 0.11 nm/cycle was maintained as the process progressed up to 40 cycles.

## 1. Introduction

Advances in semiconductor process technologies through miniaturization have resulted in their critical dimension reaching a level of less than 10 nm. However, for achieving a miniaturization of 10 nm or less, various process technologies have reached their physical limits, and new process technologies are required. Changes in the 3D structures of transistors, such as gate-all-around and fin field-effect transistors, have been rapidly implemented, and methods such as monolithic 3D and through-silicon via are replacing the existing miniaturization processes [1,2,3,4,5,6,7,8,9]. Semiconductor processing technology has started to develop in various fields, and dry etching is an important part of the semiconductor fabrication process. However, when the conventional dry etching process is used to produce micropatterns of 10 nm or less, ions and photons generated from the plasma cause damage and charged particles result in surface damage. These problems make precise etching challenging [10,11,12,13]. In particular, etching results in profile distortion in micropatterns, which is a critical problem. Accordingly, interest in atomic layer etching (ALE) technology, which is capable of atomic layer level processing, is rapidly increasing. ALE technology uniformly etches an atomic layer level thickness through a self-limited reaction [1,2,3,4,5]. It can address various etching issues, such as low etch selectivity, uniformity, and increased surface damage that occur during conventional dry etching. Additionally, because precise etching is possible at the atomic layer level, it is considered to be a suitable etching technology for fabricating next-generation semiconductors with sizes less than 10 nm.

Generally, ALE comprises an adsorption process, wherein the surface of the substrate is adsorbed with a reactive material, a desorption process, wherein the adsorbed substrate surface layer is desorbed, and purge step, wherein the residual gases that remain during the adsorption step and the desorption step are purged. In an adsorption process, the surface to be etched is modified by adsorbing a reactive gas or radicals by the formation of a chemically modified layer. The surface modified layer is then removed by ions with a controlled energy using plasma. Through a repetitive cycle process including a purge step, it is possible to remove layer by layer.

These processes increase the system cost and processing time between chambers, thereby decreasing productivity [14]. Kim et al. used an Ar ion beam and Cl_2_ gas for the ALE process, and employed a long adsorption time of 20 s or more. Although Samantha et al. reduced the adsorption time to 2 s by using a high adsorption gas pressure and power condition of 8 Pa and 900 W in silicon ALE, respectively, they failed to control the thickness at the atomic level [15,16]. Additionally, according to some research, when plasma is adsorbed during discharge in the adsorption step of the ALE process, other particles such as ions and electrons are generated in addition to radicals, which may unintentionally damage the surface. Kanarik et al. mentioned that in the adsorption step, wherein etching should not be performed, it is possible to etch using Cl^+^ and Cl_2_^+^ ions or photons; therefore, conditions that minimize this should be selected [17,18].

In this study, to address the problem of low productivity in the ALE process and the unintentional surface damage that may occur during adsorption, a mesh-grid is used in the chamber to remove ions and electrons generated via the plasma during adsorption, and only Cl radicals are selectively adsorbed. Thereafter, cyclic ALE is performed by desorbing the adsorbed layer with Ar plasma.

## 2. Experiment

### 2.1. System Setup

Figure 1a shows a schematic of the ALE system used in this study. To perform ALE in a single chamber, quartz in the vacuum chamber was fabricated into a hat shape. Cylindrical and spiral coils were installed on the upper and lower parts of the quartz, respectively. A mesh-grid was installed in the middle of the quartz and grounded to remove ions and electrons, and selectively adsorb only radicals. Figure 1b shows the details of the mesh-grid. The mesh-grid is made of stainless steel, and the size of aperture is 8 µm × 8 µm. In addition, since the mesh-grid is grounded to the chamber, ions and electrons generated in the plasma are grounded and removed. By locating a mesh-grid, the plasma is confined between the cylindrical source and the mesh-grid, and most of energetic ions in the plasma are also blocked by the mesh-grid and only low energy radicals can reach the silicon wafer surface. Owing to the mesh-grid, the plasma in the adsorption stage is discharged in the upper part, and that in the desorption stage is discharged in the lower part, that is, in the chamber. The cylindrical and spiral coils were individually connected to a 13.56 MHz radio frequency (RF) generator (Youngshin RF, Hanam, Korea, YSR-06AH) through a matching box. Additionally, the input power applied to the substrate was connected to a function generator (Hewlett Packard, Palo Alto, CA, USA, HP-8657B) and an amplifier (Electronic Navigation Industries, Nelson, New Zealand, ENI 3200L) through the matching box. Moreover, an RF of 12.56 MHz was used to prevent interference with the 13.56 MHz RF of the source. Cl_2_ was used as the process gas during adsorption, whereas Ar gas was used during desorption. Each gas was supplied through a mass flow controller. Cl_2_ was supplied to the upper part, and Ar was supplied through the shower head inside the chamber. To prevent temperature change in the substrate owing to plasma during the experiment, it was maintained at 10 °C using a chiller. To measure thickness according to ALE, a silicon on insulator wafer (substrate silicon, oxide 100 nm, poly silicon 500 nm) was used, and the change in thickness was measured using an ellipsometer (Film Sense, Lincoln, NE, USA, FS-1).

### 2.2. Analysis Tool

In this study, a home-made ion saturation current probe was used to evaluate whether only radicals can be selectively adsorbed by removing ions and electrons through the mesh-grid. The probe system, which consists of 12 tips located 10 mm above the substrate and 150 mm below the quartz, was biased to −60 V for detection of positive ions. An ion saturation current probe was inserted into the chamber, and RF power was applied to each coil to discharge plasma. Thereafter, the ion current was measured while varying the RF power and gas pressure. The process window was also checked by measuring thickness before and after the ALE process, and an ellipsometer was used to measure the accurate etch per cycle (EPC). By measuring the EPC according to the bias voltage change, sputtering is checked in the starting process conditions.

Additionally, changes in EPC were observed after each adsorption and desorption step, and also when they were alternated. Finally, as the adsorption time decreased, ALE was performed and EPC was measured to perform an optimization study according to the reduction in adsorption time.

## 3. Results and Discussion

Figure 2 shows the ion saturation current measured under various process conditions using the proposed adsorption and desorption modules. Ion saturation current was measured while changing the input power to the spiral inductively coupled plasma (ICP) coil shown in the upper part of the figure from 30 to 50 W, and changing the process pressure from 0.26 to 0.093 Pa. When the input power changes, the process pressure is fixed at 0.66 Pa, and when the process pressure changes, the input power is fixed at 50 W. As shown in the figure, the ion saturation current increased as the applied input power was increased, and measured relatively higher at the center than at the edge of the substrate. Similarly, the ion saturation current also changed as the process pressure was changed. 

For the adsorption step, reactive radicals can be adsorbed on the surface by three different adsorption mechanisms, such as reversible saturation, irreversible non-saturation, and irreversible saturation. Reversible saturation means a physisorption mechanism in which the reactive radicals are adsorbed with a weak bond of van der Waals force on the substrate surface. In the case of adsorption by reversible saturation, when the adsorption gas flow is stopped, the amount of the adsorbed species on the substrate surface is gradually decreased by the desorption of the adsorbed radicals. Irreversible non-saturation is another physisorption mechanism having thicker deposition of the reactive gas species, without saturation on the substrate surface. In this case, when the reactive gas flow is stopped, no desorption of adsorbed species is observed. Irreversible saturation is obtained when the reactive gas or radicals form strong chemical bonds with the atoms on the substrate surface. The reactive gas molecules/radicals are saturated by one monolayer on the substrate surface and, even though the reactive gas flow is stopped, the chemisorbed reactive gas or radicals on the surface remain without desorption [1]. The purpose of this study is to induce chemisorption by using Cl gas with high electronegativity and proceed with atomic layer etching without surface damage. Therefore, removing energetic ions and extracting only pure radicals using this module is a very important part. In the case of the cylindrical ICP coil shown below, no ion saturation current was observed, even when the input power and process pressure were changed. When plasma is discharged by applying input power to the cylindrical ICP coil, no ion saturation current can be observed, probably because charged particles, such as ions or electrons, generated by the plasma are grounded and removed through the mesh-grid. In general, ions with high energy may damage the Si surface by generating an ion bombardment effect. In addition, the surface roughness after etching is not good, and it is difficult to control the thickness of the atomic layer through the surface saturation reaction. These results, however, indicate that the proposed adsorption module used can perform low-damage ALE through surface reaction because the charged particles mentioned in previous studies are not generated.

In order to control the atomic level by inducing a self-limited reaction through surface adsorption, it is considered important to perform the ALE process in a region where sputtering does not occur. Generally, the ALE process controls the layers through the adsorption and desorption processes. Figure 3 shows the sputtering per cycle results according to the change in bias voltage, through which the bias voltage conditions for the desorption process can be identified. For the reliability of the experiment, this is the result measured after repeating the process for 30 cycles at each bias voltage. As shown in the figure, the Si layer was not sputtered when the bias voltage was lower than 7 V, whereas it was sputtered when the bias voltage exceeded 7 V. This indicates that the Ar ions were not sufficiently energetic to break the Si–Si bond when the bias voltage was lower than 7 V; however, once it exceeded 7 V, they could break the Si–Si bond. Therefore, the ALE desorption condition was set to 7 V, which was the threshold energy at which the Si–Si bond was not broken and a synergistic effect could be obtained during adsorption. It does not mean that the synergistic effect by adsorption does not occur when the bias voltage is higher than 7 V. If the ion energy is higher than 7 V (the sputter threshold energy), the sputtering per cycle is noticeably increased with the increase in ion energy by sputter, etching the atoms located under the adsorption layer after the desorption process. Therefore, there is an appropriate bias voltage range for ions, and it is important to control it precisely according to the materials to be etched. 

Figure 4 shows the EPC results when only adsorption or desorption was performed, and when both adsorption and desorption were performed as a cyclic process. Desorption was performed under 200 W of source power, 7 V of bias voltage, and 0.66 Pa of Ar gas. Adsorption was performed for 30 s under 0.4 Pa of Cl_2_ gas and 30 W of source power. As shown in the figure, when only adsorption or desorption was performed, the Si substrate was not etched or etched finely, but when both adsorption and desorption were performed as a cyclic process, it was etched at a rate of approximately 0.11 nm/cycle. Generally, when Cl radicals generated through adsorption are adsorbed into the Si substrate, Si–Cl bonds form on the surface, and the lower Si–Si bond energy is weakly changed. This indicates that only the radical-adsorbed layer is etched, and if this is continuously performed through a cyclic process, a layer etch rate of approximately 0.11 nm/cycle can be obtained [19,20]. Based on the above results, the meaning of the ideal ALE process can be extended to the etching selectivity. If the self-limited reaction on the surface through the adsorption step and the amount of sputtering can be reduced close to zero, it is considered that it will have a significant positive effect in terms of the etching selectivity because only the adsorption layer can be selectively removed.

In general, the ALE process consists of adsorption, desorption, and purge steps, so it takes a lot of time and has been reported to have poor productivity. The ALE process through the self-limited reaction eliminates the effects of transport phenomena, a major cause of problems such as aspect ratio dependent etching (ARDE). In addition, it has a positive effect on the roughness of the surface, which is considered important in the subsequent process. In this study, ALE characteristics were studied by improving the adsorption step to shorten the process time. Figure 5 shows the EPC results after the ALE process as the adsorption time was increased up to 30 s. As can be seen in the figure, at adsorption times from 5 s to 30 s, it was observed that EPC hardly changed or slightly decreased. It was found that when the adsorption time was set to 5 s or longer, the Cl radical was sufficiently adsorbed on the sample surface and did not affect the self-limited reaction anymore and did not affect the EPC. However, when the adsorption time was reduced to 3 s, it was confirmed that the EPC slightly decreased to about 0.1 nm/cycle. It is considered that the EPC was reduced because the adsorption time was too short and Cl radical was partially adsorbed on the sample surface. This means that sufficient radicals were not generated in the adsorption module, and it is considered that the adsorption time can be further shortened if high-density plasma can be generated at high process pressure and input power.

In general, the ALE process takes a lot of time as many cycles are repeated, and temperature rises through collision from plasma active species in the substrate, and process reliability resulting from this, are mentioned as important issues. Figure 6 shows the etch depths and EPC results obtained by changing the ALE process up to 40 cycles. The desorption was performed under 200 W of source power, 7 V of bias voltage, and 0.66 Pa of Ar gas. The adsorption was performed for 5 s under 4 Pa of Cl_2_ gas and 30 W of source power. As shown in the figure, the etch depth increased linearly as the number of cycles increased. Additionally, a similar EPC of approximately 0.11 nm/cycle was observed for different EPC cycles. This result showed that the bias voltage applied to the substrate and the substrate temperature were stably maintained as the process progressed up to 40 cycles. In particular, it is considered that the result shows the reliability of the adsorption module applied to improve productivity.

## 4. Conclusions

In this study, an ALE process using the proposed adsorption module was evaluated. In order to control the atomic level by inducing a self-limited reaction through surface adsorption, threshold ion energy region was investigated. If the ion energy is higher than 7 V (the sputter threshold energy), the sputtering per cycle is noticeably increased. To reduce the adsorption time and damage caused by charged particles, such as ions or electrons, generated by the plasma during adsorption, an adsorption module that can extract only radicals was fabricated.

Using the fabricated adsorption module, the adsorption time was reduced from 30 to 5 s, and an EPC of approximately 0.11 nm/cycle was maintained. Additionally, etch depth and EPC were observed during the ALE process up to 40 cycles to verify the reproducibility of the process. As the number of ALE cycles increased, the etch depth increased linearly, and a similar EPC of approximately 0.11 nm/cycle was maintained.

## Figures and Tables

**Figure 1 materials-16-03611-f001:**
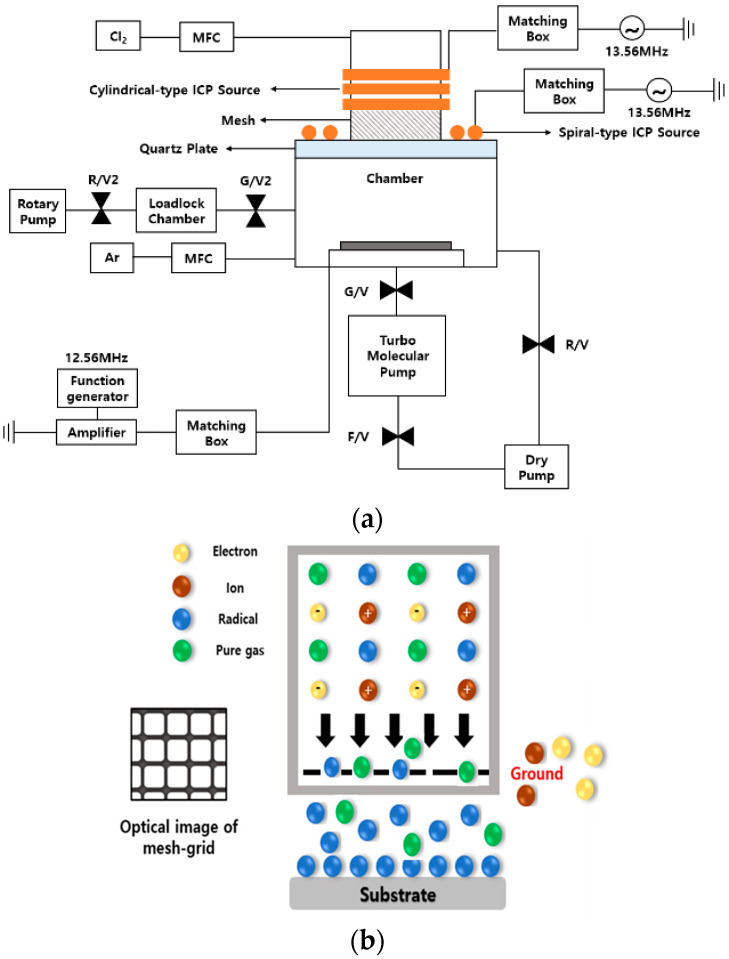
(**a**) Schematic of the ALE system with the proposed adsorption module (**b**) Details of the mesh-grid.

**Figure 2 materials-16-03611-f002:**
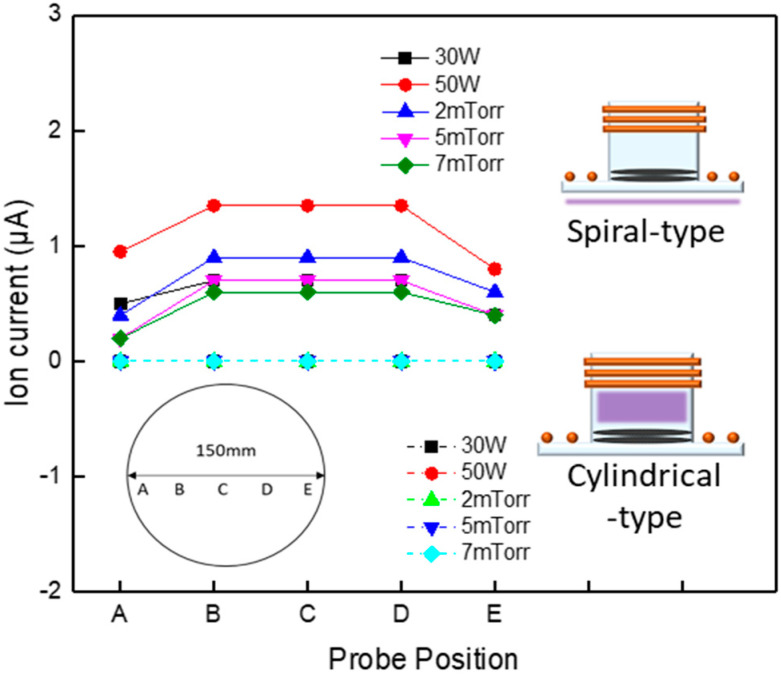
Ion saturation current measurements according to input power and process pressure change at points (A–E) marked on the substrate of 150mm in the adsorption module.

**Figure 3 materials-16-03611-f003:**
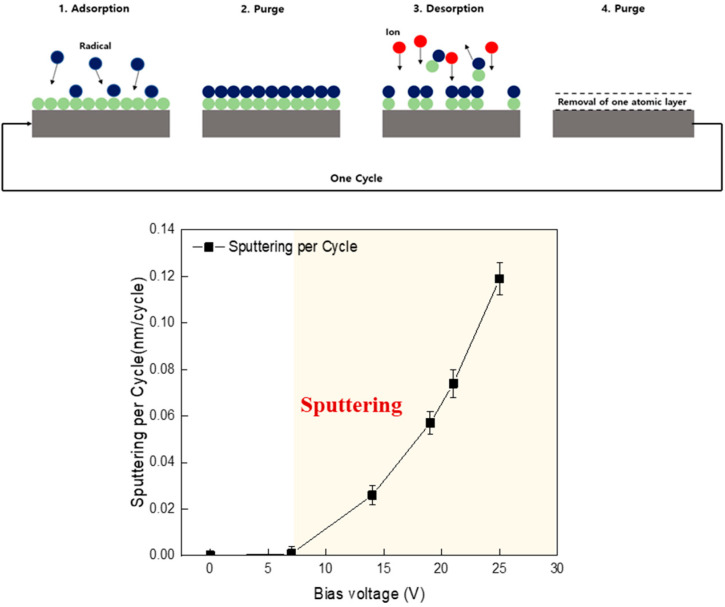
Sputtering per cycle according to bias voltage change in the ALE system.

**Figure 4 materials-16-03611-f004:**
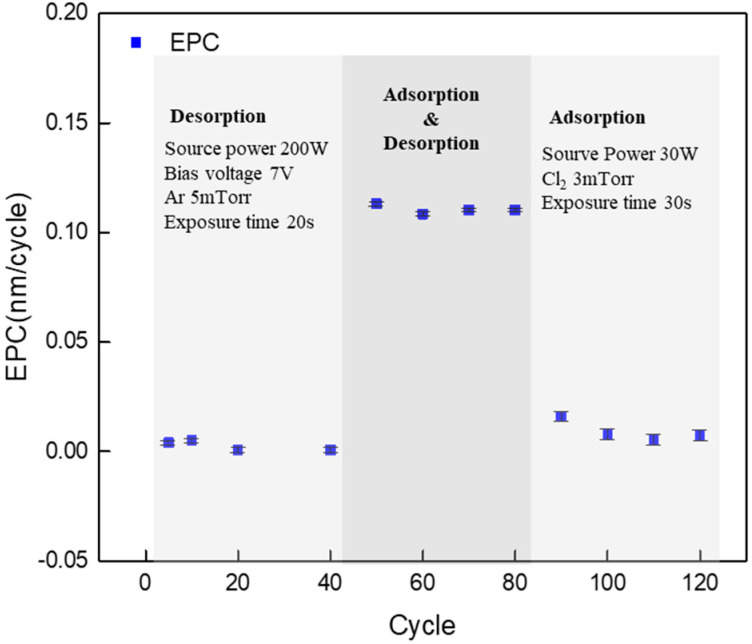
EPC change during each adsorption and desorption process, and in the cyclic process including both adsorption and desorption.

**Figure 5 materials-16-03611-f005:**
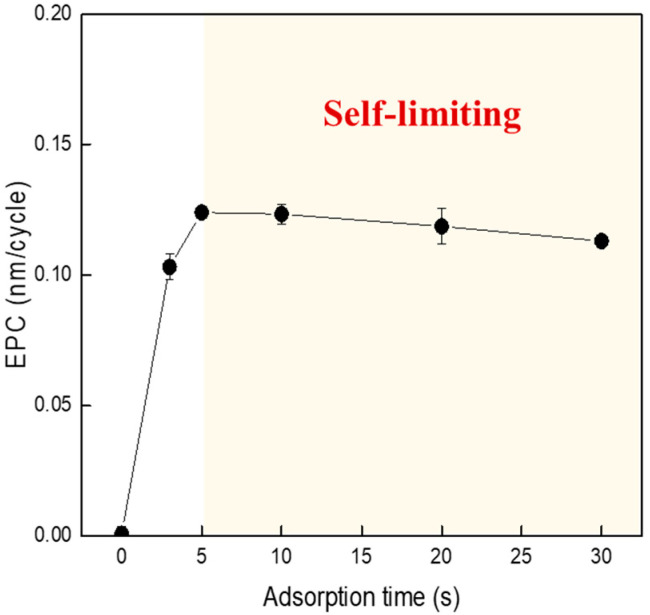
EPC according to adsorption time change in the ALE process.

**Figure 6 materials-16-03611-f006:**
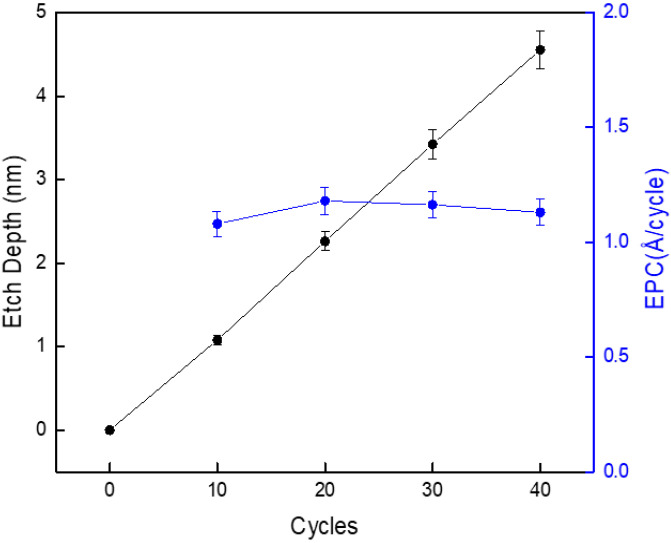
Etch depth and EPC for the ALE process up to 40 cycles.

## Data Availability

Not applicable.

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
