# Peer review of "Atomic Layer Etching Using a Novel Radical Generation Module"

_materials, 2023, doi:10.3390/ma16103611_

Round 1

Reviewer 1 Report

This manuscript describes a method to selectively utilize low energy radical ions for ALE process in order to improve the productivity and reduce the surface damage during absorption phase. The authors thoroughly demonstrated the experiment setup (both etching and measurement) and result analysis. The results presented in this manuscript match with the proposal and expectation while there are some minor revisions needed before publication:

1. The key part of the module is the mesh-grid. Authors shall explain more on its design. What is the size of aperture for the mesh-grid in order to selectively let radical ions pass and be absorbed? What is the material used to fabricate it? 

2. In Figure 2, authors shall elaborate more on the legends. What is the pressure condition when adjusting the input power? What is the input power when adjusting the process pressure?

3. This manuscript mainly demonstrated the etching process of poly-Si on insulator by using Cl radical. Is there any extended study on etching other materials such as Al2O3 by using a different radical? Would this module still work?

Author Response

This manuscript describes a method to selectively utilize low energy radical ions for ALE process in order to improve the productivity and reduce the surface damage during absorption phase. The authors thoroughly demonstrated the experiment setup (both etching and measurement) and result analysis. The results presented in this manuscript match with the proposal and expectation while there are some minor revisions needed before publication:

Point 1 : The key part of the module is the mesh-grid. Authors shall explain more on its design. What is the size of aperture for the mesh-grid in order to selectively let radical ions pass and be absorbed? What is the material used to fabricate it? 

Response 1 : The mesh grid is made of stainless steel, and the size of aperture is 8x8µm2. As the reviewer pointed out, the relevant parts have been corrected and supplemented.

Point 2 : In Figure 2, authors shall elaborate more on the legends. What is the pressure condition when adjusting the input power? What is the input power when adjusting the process pressure?

Response 2: When the input power was changed from 30W to 50W, the process pressure was fixed at 5mTorr. Also, when the process pressure changed from 2 mTorr to 7 mTorr, the input power was fixed at 50 W. As the reviewer pointed out, the relevant parts were corrected and supplemented.

Point 3 : This manuscript mainly demonstrated the etching process of poly-Si on insulator by using Cl radical. Is there any extended study on etching other materials such as Al2O3 by using a different radical? Would this module still work?

Response 3: Currently, there are no atomic layer etching research results for Al2O3 that we have conducted. However, since there are many etching research results using chlorine gas, it is considered that it can be fully applied and studied.

Reviewer 2 Report

The publication is easy to read and understand in large parts. However, I have some comments and questions:

Most important point:
I do not understand Fig2 with the description (lines 109-115). What pressures are set for the processes 30 W and 50 W, what powers for the different pressures? I can't make sense of the results without these parameters.

In general, pressures should be given in the SI unit (Pa or hPa), even if the machine parameters specify mTorr. I find even angstroms exhausting as I convert it to nm.

Other questions regarding the study that should be explained in the paper:
How thick is a monolayer/atom layer ?  I expect values for Si here to be on the order of 0.4 nm, so is 1.1 Angstrom just an averaging and not an atomic layer? Are there any assumptions regarding the thickness of one Cl2-Si layer? Is the rate Angstrom/Cylcle determined in each case from the etch depth after etching. A more detailed explanation of the layer thickness measurement would also be beneficial. I assume this was done after end of the etch process, or is an insitu measurement also possible?
Is the substrate temperature measured during etching, if so how, or is the temperature here a machine parameter? Can "fast" etching continue even with the changes to the reactor chamber, or is the equipment used for ALE alone?

Author Response

The publication is easy to read and understand in large parts. However, I have some comments and questions:

Point 1 : I do not understand Fig2 with the description (lines 109-115). What pressures are set for the processes 30 W and 50 W, what powers for the different pressures? I can't make sense of the results without these parameters.

Response 1: When the input power was changed from 30W to 50W, the process pressure was fixed at 5mTorr. Also, when the process pressure changed from 2 mTorr to 7 mTorr, the input power was fixed at 50 W. As the reviewer pointed out, the relevant parts were corrected and supplemented.

Point 2 : In general, pressures should be given in the SI unit (Pa or hPa), even if the machine parameters specify mTorr. I find even angstroms exhausting as I convert it to nm.

Response 2: As reviewer commented, the MKS unit used in the paper was modified to the SI unit.

Point 3 : How thick is a monolayer/atom layer ?  I expect values for Si here to be on the order of 0.4 nm, so is 1.1 Angstrom just an averaging and not an atomic layer? Are there any assumptions regarding the thickness of one Cl2-Si layer? Is the rate Angstrom/Cylcle determined in each case from the etch depth after etching. A more detailed explanation of the layer thickness measurement would also be beneficial. I assume this was done after end of the etch process, or is an insitu measurement also possible?

Response 3: For reference, the interatomic distance of (1,0,0) Si layers is 0.136 nm. With the simplest case of Si attaching to one Cl and rearranging to the desorption product SiCl2, theoretically, EPC = 0.068 nm. This is consistent with experimentally observed EPC = 0.067 nm in thermal Si ALE with Cl2 gas. Our ALE experiment results showed a higher EPC of about 0.1 nm. This is slightly different from the theoretical value, and it is considered possible if the sputtering conditions are optimized. As the reviewer points out, the EPC observed in this study is considered to be an average Si EPC for cyclic etching. EPC (etch per cycle) can be measured in real time using an ellipsometer installed in the chamber. However, since there is a slight deviation in the amount of etching for each cycle, the average EPC (etch per cycle) is measured by performing 40 cycles.

Point 4 : Is the substrate temperature measured during etching, if so how, or is the temperature here a machine parameter? Can "fast" etching continue even with the changes to the reactor chamber, or is the equipment used for ALE alone?

Response 4: The temperature of the substrate was measured using a thermocouple sensor, and due to the nature of the ALE process, the temperature does not increase much as the cycle repeats. The system used in this study was developed as an ICP etcher and was modified to perform the cyclic etch process for the ALE process. Therefore, both conventional fast etching and atomic layer etching can be performed using one chamber.

Reviewer 3 Report

This paper describes an improvement idea of radical supply in ALE. This paper will provide interesting experimental results for many readers. However, from a technical point of view, the description is insufficient. The paper needs to be revised in consideration of the following points.

>Fig.1

The point of this research is the effect of "mesh-grid". Details of the "mesh-grid", including size, should be illustrated in Figure 1 to aid the reader's understanding. The ground connection for the "mesh-grid" also needs to be clarified. The mechanism of removal of ions and electrons should also be explained using figures. For example, Fig.1(a) should be the already described schematic of ALE system, and Fig.1(b) should be added to explain the details of the "mesh-grid" and the mechanism of removing ions and electrons. (The text also needs to be added).

>Fig.2

The details of the probe point positions (A, B,...) are not described. These are also not mentioned in the text. It is necessary to add an explanation in the text as well as an illustration.

>Line 156 This indicates that the ions were not....

Are the ions described here Ar ions? If so, please add this.

>Fig.4

The RF power for generating Ar plasma in the desorption step is 200 W. This value indicates a relatively large power. How many "mm" thick is the sheath at this time? Please explain the relationship between the sheath thickness and the bias voltage of 7V.

>It is recommended to add an example of observation of ALE on the sidewall of a fine trench structure using this experimental system.

Author Response

This paper describes an improvement idea of radical supply in ALE. This paper will provide interesting experimental results for many readers. However, from a technical point of view, the description is insufficient. The paper needs to be revised in consideration of the following points. 

Point 1 : >Fig.1

The point of this research is the effect of "mesh-grid". Details of the "mesh-grid", including size, should be illustrated in Figure 1 to aid the reader's understanding. The ground connection for the "mesh-grid" also needs to be clarified. The mechanism of removal of ions and electrons should also be explained using figures. For example, Fig.1(a) should be the already described schematic of ALE system, and Fig.1(b) should be added to explain the details of the "mesh-grid" and the mechanism of removing ions and electrons. (The text also needs to be added).

 Response 1: The mesh grid is made of stainless steel, and the size of aperture is 8x8µm2. As the reviewer pointed out, the relevant parts have been corrected and supplemented.

Point 2 : >Fig.2

The details of the probe point positions (A, B,...) are not described. These are also not mentioned in the text. It is necessary to add an explanation in the text as well as an illustration.

 Response 2: As reviewer commented, the figure related to the probe position measured on the substrate were corrected and supplemented.

Point 3 : >Line 156 This indicates that the ions were not....

Are the ions described here Ar ions? If so, please add this.

 Response 3: As suggested by the reviewer, we have clarified the relevant wording. 

Point 4 : The RF power for generating Ar plasma in the desorption step is 200 W. This value indicates a relatively large power. How many "mm" thick is the sheath at this time? Please explain the relationship between the sheath thickness and the bias voltage of 7V.

Response 4: As suggested by the reviewer, we have clarified the relevant wording. In general, as the input power applied to the ICP increases, the high energy position in the ion energy distribution moves to the low energy region. Also, as the sheath potential decreases, the self-bias voltage applied to the substrate decreases. The substrate voltage of 7V used in this study is the value obtained by applying 12.56MHz to the substate for independent ion energy control.

Point 5 : >It is recommended to add an example of observation of ALE on the sidewall of a fine trench structure using this experimental system.

Response 5: The adsorption module is being optimized for the entire substrate. There are no remarkable results using pattern sample, so it is difficult to add research on this. However, if all research is optimized and finished, we will have an opportunity to report on this again.

Round 2

Reviewer 1 Report

Authors have addressed my questions appropriately and corresponding editions have been done in the manuscript. 

Reviewer 2 Report

The submitted manuscript has been sufficiently revised.

Reviewer 3 Report

The submitted manuscript has been sufficiently revised.